# Modeling of Multiple Fatigue Cracks for the Aircraft Wing Corner Box Based on Non-ordinary State-based Peridynamics

Junzhao Han [1] , Guozhong Wang [2], Xiaoyu Zhao [3], Rong Chen [1] and Wenhua Chen [1,*]

1 School of Mechanical Engineering and Automation, Zhejiang Sci-Tech University, Hangzhou 310018, China; hanjz@zju.edu.cn (J.H.); rchen@zstu.edu.cn (R.C.)
2 Henghong Intelligent Equipment Co., Ltd., Hangzhou 310018, China; gzwang@hzjx.com.cn
3 China Ship Scientific Research Center, Wuxi 214028, China; zhxiyu123@126.com
* Correspondence: chenwh8@zju.edu.cn

**Abstract:** In the current research, we propose a novel non-ordinary state-based peridynamics (PD) fatigue model for multiple cracks' initiation and growth under tension–tension fatigue load. In each loading cycle, the fatigue loading is redistributed throughout the peridynamic solid body, leading to progressive fatigue damage formation and expansion in an autonomous fashion. The proposed fatigue model parameters are first verified by a 3D numerical solution, and then, the novel model is used to depict the widespread fatigue damage evolution of the aircraft wing corner box. The modified constitutive damage model has been implemented into the peridynamic framework. Furthermore, the criteria and processes from multiple initiations to propagation are discussed in detail. It was found that the computational results obtained from the PD fatigue model were consistent with those from the test data. The angular errors of multiple cracks are within 2.66% and the number of cycles errors are within 15%. A comparison of test data and computational results indicates that the fatigue model can successfully capture multiple crack formations and propagation, and other behaviors of aluminum alloy material.

**Keywords:** non-ordinary state-based peridynamics; tension–tension fatigue load; multiple cracks; aircraft wing corner box

## 1. Introduction

Extensive fatigue damage of aging aircraft structures has aroused widespread attention in the aerospace field. The prediction of the fatigue initiation and propagation of a structure is complicated, tough work because of the uncertain loads and uncertainties in material properties [1–3]. The fatigue damage law should be deduced to ensure flight safety and to improve costs. After in-depth research, it was found that fatigue is the formation and expansion of damage in a material under cyclic loads. Two stages can depict the whole process based on the cumulative damage theory. The first stage is the crack formation, during which damage nucleates in different locations of the material, and it can be impacted by various mechanical, microstructural, and environmental factors. The second stage is crack propagation, during which fatigue damage propagates stably until it exceeds the safety margin, followed by rapid propagation that leads to catastrophic fracture [4–7]. Therefore, the fatigue of metallic structures is a complex, irreversible process under repeated loading, in which damage formation is controlled by the interaction of micro-cracks, trans-scale effects on geometry and time, and unpredictable factors [8–11]. Vacancies are generated when an atom or an ion is missing from its regular crystallographic site and new equilibriums are built. Under repeated loading, the new dynamic equilibriums are lost and generate new balances, and fatigue damage initiates and accumulates naturally [11–13]. Therefore, the metal fatigue process includes complex submicroscopic and microscopic damage propagation [14–16]. The fatigue life of metal structures under cyclic loading can be accurately predicted from crack initiation to propagation, which is influenced by the

material properties, geometric parameters, boundary loads, external environment, etc. The whole process of fatigue is essentially a complex, multi-scale phenomenon that has important features at multiple scales of time and/or space [17–21]. Therefore, the prediction of fatigue damage in metallic materials encounters many problems [22–24]. The research hypothesis of the classical continuum local theory is that a specified material particle contacts other points within a local range. Out of the range, there is no force effect between the two particles. Within the realm of the above hypothesis, the motion of a considered particle is controlled by spatial partial differential equations leading to the stress at the crack tip being mathematically singular [25,26]. Therefore, the crack formation and expansion are simulated separately with extra criteria. The enriched finite-element and many other modified damage models focus on the work of building the relationships between damage parameters at the fatigue crack tip which characterizes the damage evolution under cyclic loading [27–29]. The main barrier is that these models cannot be directly used on the fatigue crack tips or crack surfaces. The existing methods take remedial measures, such as redefining the body in every calculating procedure and adding additional equations, which means that the extra judging criteria to guide the crack path and angle are deduced [30–32]. To deal with such issues, the stage breakthrough in the computational mechanics is introducing the traction–separation law for the opening model I fracture mechanism and intermix model under external loads. Based on the presented law, the interfaces in the material are modeled and cohesive zone elements are placed on the undefined region in the classical methods. Fatigue crack growth shows the mesh dependence property, and a special remeshing technique is required. However, the crack path and angle are sensitive to grid feature and density. Frequently remeshing the solid body is complicated and the result is difficult to converge for simple crack growth modeling, let alone the multiple crack propagation model. To overcome those difficulties, the eXtended Finite Element Method is proposed as a compromise to avoid remeshing in the fatigue crack propagation process. Local enhancement function and permitting the crack to propagate on element surface are its typical characteristics. Neither the FEM model, various modified versions, nor the XFEM method suffers from the need to supply more control equations, such as the location of the fatigue crack nucleate, crack angle and direction, crack expansion, and arrest during the external loading. In other words, fatigue crack formation and expansion use a single-handed treatment scheme coupling dissimilar mathematical systems under the framework of traditional continuum theory. The whole process of fatigue fracture is directly controlled by dislocations and grain boundary, where the microcracks are a priority to initiate in those locations [33–37]. It crosses space and time scales from small cracks to macroscopically visible cracks. Therefore, there are difficulties predicting the site of fatigue nucleation with a numerical solution, and it is also difficult to obtain such a conclusion from a complex and changeable test process [38–43].

Although many laws and equations are deduced to depict the two stages in the cracks and fissures evolution, the key to the crack problem is still unresolved and complicated within the realm of classical continuum local theory. The crux and contradicting step in the framework is applying the equation of continuity to the discontinuity in the body leading the irreconcilable results. When cracks or discontinuities appear, the control equation of the material structure fails to deal with these processes, and many remedial treatments are proposed to strip discontinuity out of the initial framework. This operation process promotes fuzziness and a lack of clarity, especially with the nucleation and expansion of cracks [44]. Classical numerical procedures such as FEM and XFEM methods involve a high dependence on element size and boundary treatment. To reduce that dependence and obtain accurate results, a meshless method was developed based on continuum mechanics. The new material damage model satisfies two rules [45]. First, accurate numerical solutions in the crack tip are rational without singular stress and strain value, where the transition from the continuous phase to the discontinuous phase occurs smoothly. Second, the fracture model and crack path are highly consistent with experimental results observed in the experimental laboratory. A new nonlocal framework called peridynamics was proposed

by Silling in 2000 based on the inadequacies of local and nonlocal algorithms. The essence of this nonlocal theory is the reconstruction of particle control equations that use the new model to depict damage evolution in a peridynamic solid. The given particle contacts with other neighbors in a finite range called the peridynamic radius and integral-type representation is better at solving the discontinuity problem than in partial differential forms. Peridynamics exhibited advantages in dealing with characteristic crack prediction, which can be carried out to simulate fatigue damage during cyclic loading [46–48]. Material damage initiates and propagates naturally; thus, extra criteria are unnecessary. Additionally, the path of the crack is arbitrary due to the novel particle interactions among the peridynamic solid, which is different from the classical framework that only propagates along the boundary of the finite element. Furthermore, the PD damage model comes across lengths of multiple scales, from micro to macro, during the fatigue loading [49–51]. This formulation allows us to model crack initiation, propagation, branching, and coalescence without special assumptions.

The peridynamic fatigue damage law was first presented by Oterkus and Madenci [52], and then Askari [53] proposed a modified version based on basic physical theory. The whole process of fatigue damage within the cyclic loading was implemented in a continuous model, and the fatigue characteristic parameters were verified with classical fatigue test results. In [53], the bond damage model deals with fatigue crack initiation and propagation arise applied to cyclic loading. A suitable damage law is deduced based on the S-N curve and Paris law, leading to crack emergence and accumulation. Zhang [54–57] proposed a progressive damage prediction for fiber-reinforced composites based on the peridynamic theory to predict strength and failure progression. The test data show that the reduction in stiffness and strength is good agreement with the peridynamic simulation results. The results indicate that the ordinary peridynamic fatigue damage model can deal with the details of crack initiation and growth without extra rules [58–60].

In the present investigation, a novel state-based PD fatigue model is implemented to calculate tension–tension loads. In each loading cycle, the fatigue loading is redistributed in the peridynamic solid body, leading to progressive fatigue damage formation and propagation in an autonomous fashion. The modified constitutive damage model has been implemented in the peridynamic framework, and the multiple cracks' initiation and propagation occur during the tension–tension loading. Furthermore, the criterion and process from multiple initiations to propagation are discussed in detail. The proposed fatigue model parameters are firstly verified by a 3D numerical solution, and then the novel model is used to describe the widespread fatigue damage evolution of the aircraft wing corner box. It is found that the computational results obtained from the PD fatigue model are consistent with those from the test data. A comparison of test data and computational results indicates that the fatigue model can successfully capture the multiple cracks' formation and propagation and other behaviors of the aluminum alloy.

## 2. Progressive Fatigue Damage Evolution

### 2.1. Microscopic Fatigue Crack Initiation and Propagation

Based on crystallographic theory, cumulative damage mechanisms in the materials are directly related to atomic evolution during the cyclic loading. Atoms vibrate in high frequency (about $10^{12} \sim 10^{13}$ Hz), and the small-amplitude surrounding equilibrium position. Every atom in the solid body has a certain energy which keeps it in a dynamic balance state. When the dynamic balance state of an atom is broken, the atom will jump out of its original balance site, and a void forms in the cluster of matter points. The probability of a material point energy reaching the critical value $Q_c$ because of thermal disturbance motion can be expressed as

$$P_c = \exp(-Q_c/kT) \tag{1}$$

where $k$ is a physical Boltzmann constant with respect to temperature and energy, $T$ is the representation of the average kinetic energy of an atom, and $Q_c$ is the critical value that is the smallest active energy of an atom flounce away from the balanced state.

As can be seen in Figure 1a, compared with the other two loading states, the critical active energy $Q_c$ is larger under the no external loading. There are still some atoms leaving the original equilibrium position and entering the new dynamic vacant equilibrium position. The escaping probability of the atom in all directions is the same. Therefore, in such a loading condition, no fixed crystal defects will be formed and damage evolution will be negligible.

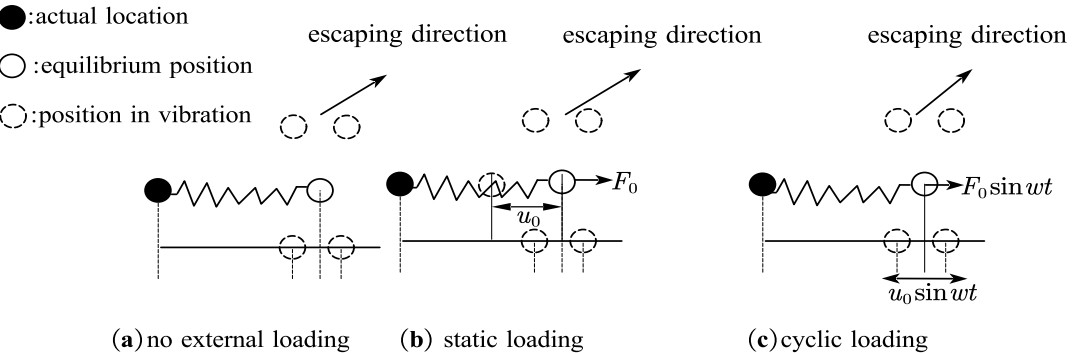

**(a)** no external loading      **(b)** static loading      **(c)** cyclic loading

**Figure 1.** Thermal perturbation in material atoms under three different load conditions.

As shown in Figure 1b, under static load, the critical active energy $Q_c$ is inversely proportional to the stress level. The higher the stress, the smaller the critical active energy is. Therefore, the atom has a higher probability of having energy above this critical active energy. Inspired by [59], and above theory, a unified model for the critical value $Q_c$ can be expressed as

$$Q_c = \frac{\tau_b^2 + \alpha \tau_b \tau + \beta \tau^2}{2G} \tag{2}$$

where $\tau_b$ is the shear strength under ideal conditions for a metallic material, $\tau$ is the external shear stress, $G$ is the shear modulus related to shear stress $\tau$, $\alpha$ is a fundamental physical parameter directly related to the coefficient of the stress concentration, $\beta$ is a fundamental physical parameter directly related to the loading amplitude, and $\alpha \tau_b \tau + \beta \tau^2 \le 0$. When $\tau = 0$, the critical active energy $Q_c$, as shown in Figure 1a, can be expressed as

$$Q_c = \tau_b^2 / 2G \tag{3}$$

As shown in Figure 1b. the displacement $u_0$ of a material point is flounced away, and the balanced state is fixed at this time, avoiding the probability of the atom in all directions being the same with respect to the new equilibrium position. Therefore, the vacancies generated by an atomic escape because of the thermal disturbance can be annihilated. Vacancies generated by atomic escape show mutual annihilation without accumulating to form defects or making defects grow. Therefore, there is no damage evolution formed under a static load.

Combining Equations (1) and (2), the possible probability of the occurrence of an atom's energy of activation reaching the critical value $Q_c$ under a static load can be expressed as

$$P_c = \exp\left(-\frac{\tau_b^2 + \alpha \tau_b \tau + \beta \tau^2}{2GkT}\right) \tag{4}$$

where the probability for a material particle under static loading to flounce away the balance state is larger than the condition without external loading.

As shown in Figure 1c, under cyclic loading, the equilibrium position is constantly changing during the deformation process. The displacement $u$ of a material point and the external force can be expressed as

$$\begin{cases} u = u_0 \sin wt \\ F = F_0 \sin wt \end{cases} \tag{5}$$

where $u$ is the displacement of an atom oscillation surrounding equilibrium position, $F$ is the cyclic load, and $w$ is the angular frequency.

The atomic escaping direction is isotropic with respect to the instantaneous equilibrium position. However, the motion of the equilibrium position itself will destroy this isotropic property. The displacement for a material particle flounces away when the balanced state is alternated. Vacancies generated by atomic escape show no annihilation, leading to the fatigue damage being initiated and accumulated.

As shown in Figure 2, under cyclic loading, fatigue damage evolution preferentially forms at grain boundaries. Micro-damage initiates at grain boundaries ($GB_1$, $GB_2$, and $GB_3$) because of continuous external loading on the load position. Multiple cracks' nucleation and expansion are a consequence cyclic slip along grain boundaries (lines in green). The whole process undergoes cyclic plastic deformation and dislocation activities. Progressive damage models are often based on damage mechanics concepts. This provides a way to set up the damage evolution law with a theoretical foundation. The evolution law based on this mechanism can indicate metals' already-known damage–life relationships.

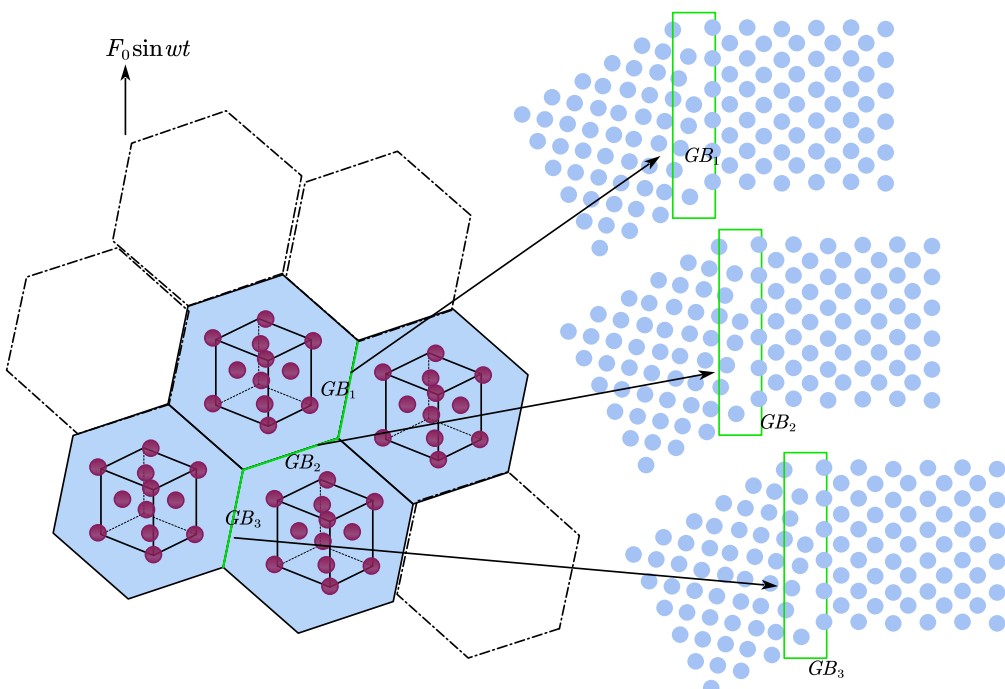

**Figure 2.** Grain boundary damage indued by cyclic loading.

## 2.2. Widespread Fatigue Damage

As shown in Figure 3a, in one loading cycle, the hysteresis loop area can be generally described as follows:

$$\Delta W = \iint d\sigma d\varepsilon_p = \oint \sigma(\varepsilon_p)d\varepsilon_p \tag{6}$$

where $\Delta W$ is the loop area, $\sigma$ is the alternating stress, and $\varepsilon_p$ is the deformation during one loading cycle. This area depends on the hysteresis-loop shape and on the stress-and plastic-strain amplitudes, which are here limits of integration.

As shown in Figure 3b, oblique shadows represent the hysteresis area equal to the energy-dissipating pile of deformation in one loading cycle. The loop area can be expressed in the form

$$\Delta W = 4\sigma_h^a \varepsilon_h - 2 \int_0^{2\sigma_a^h} \varepsilon d\sigma \tag{7}$$

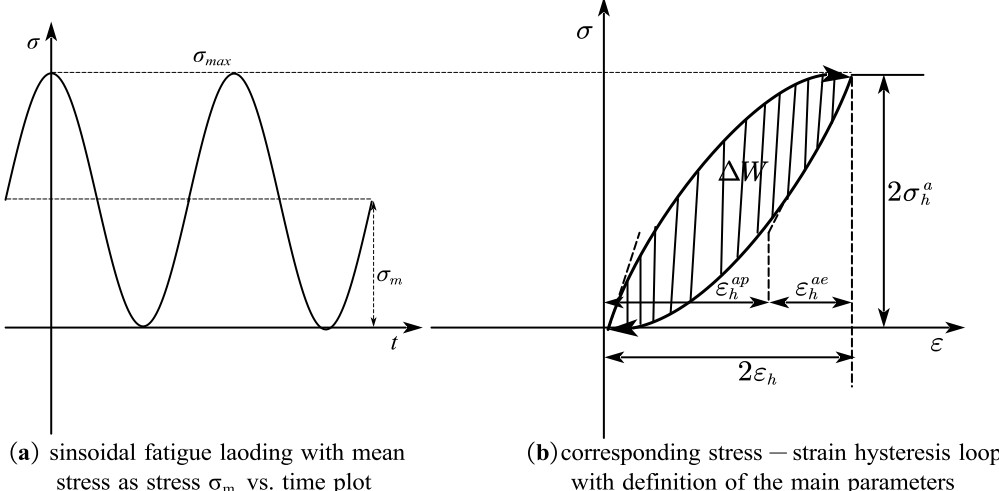

(**a**) sinsoidal fatigue laoding with mean stress as stress σ_m vs. time plot

(**b**) corresponding stress − strain hysteresis loop with definition of the main parameters

**Figure 3.** Fatigue loading spectrum and hysteresis area in one loading cycle.

The energy of the nucleus of the plastic deformation work is mainly converted into heat energy, with a small part of it converted into stored energy. If all the hysteresis loops are superposed, then the total hysteresis energy is obtained. Without considering the hardening/softening period, the total energy can be represented by the area of saturation hysteresis loop and the number of cycles to fracture. Under cyclic loading, the widespread fatigue damage can be described as

$$\sum_{i=1}^{n} \Delta S_i = \sum_{i=1}^{n} \Delta W_i^1 + \sum_{i=1}^{n} \Delta W_i^2 + \dots \tag{8}$$

where $\Delta W_i^j$ is the loop area during the $i^{\text{th}}$ cyclic loading for the $j^{\text{th}}$ crack and $\Delta S_i$ is the fatigue damage during the $i^{\text{th}}$ fatigue loading cycle.

### 2.3. Fatigue Damage Accumulation for Multiple Cracks

The cumulative damage process of materials is a nonlinear process of interaction between loading stress and damage accumulation. The three-dimℏensional damage accumulation model can be expressed as

$$\hbar_D^* = \frac{\varsigma_1}{\varsigma_2 + 1} \left(-\frac{\psi}{\varsigma_1}\right)^{\varsigma_1 + 1} \left(\dot{\omega} + \dot{c}\right) / (1 - D)^T \tag{9}$$

where $\hbar_D^*$ is the thermodynamics dissipative potential under cyclic loading conditions; $\varsigma_1$, $\varsigma_2$, and $T$ are material parameters depending on the thermodynamic temperature; $\dot{\omega}$ is the accumulative plastic strain rate; $\dot{c}$ is the microplasticity strain rate; and $\psi$ is the damaged strain energy release rate. Based on the tension–tension loading theory, the damaged strain energy release rate is described as follows:

$$\psi = -\frac{e_{eq} \varsigma_i}{2E(1 - D)^2} \tag{10}$$

where $e_{eq}$ is Huber–Mises equivalent stress, $\varsigma_i$ is the triaxial stress factor, $D$ is the damage variable, and $E$ is the elastic modulus. The triaxial stress factor can be expressed as

$$\varsigma_i = \frac{2}{3}(1 + \nu) + 3(1 - 2\nu)\left(\frac{e_H}{e_{eq}}\right)^2 \tag{11}$$

where $e_H$ is the hydrostatic pressure and $\nu$ is Poisson's ratio for a metallic material.

Based on orthogonality principles, the damage variable rate can be described as follows

$$\dot{D} = -\frac{\partial \hbar_D^*}{\partial \psi} = \left(-\frac{\psi}{\varsigma_1}\right)^{\varsigma_1}\left(\dot{\omega} + \dot{c}\right)/(1-D)^T \tag{12}$$

## 3. State-Based Peridynamics for Multiple Cracks

### 3.1. The Motion and Deformation of a Material Point

According to Figure 4a, the cumulative damage process of materials is a nonlinear process of interaction between loading stress and damage accumulation. The three-dimensional damage accumulation model is described as

$$\rho(x)\ddot{u}(x,t) = \int\limits_{H_x} \left(t(u'-u,x'-x,t) - t'(u-u',x-x',t)\right)dH_x + b(x,t) \tag{13}$$

where $\rho(x)$ means the mass density for every material point in the reference configuration, $x$ is the vector of the coordinates for a given material point, $u$ is the acceleration vector in the reference configuration, $u'$ is the displacement vector in the deformed state, and $b(x,t)$ is the vector of body forces. In the peridynamic solid body, external loading is carried out on the mathematical model. The motion of a given material particle satisfies the Lagrangian equation in the deformed configuration.

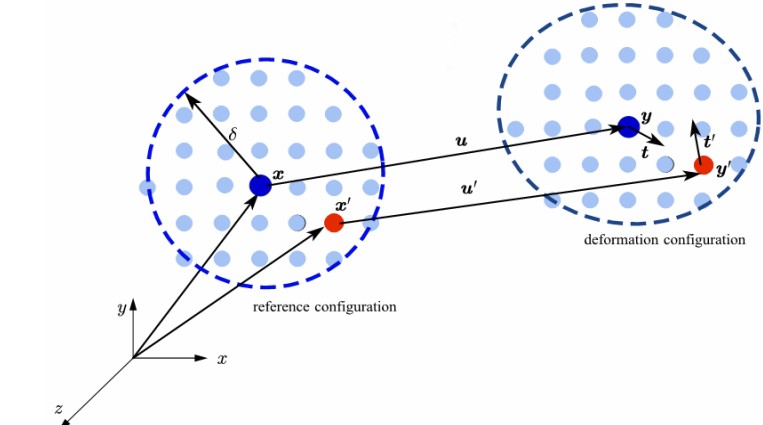

(**a**) Defromation of PD material points x and x' and developing unequal pairwise force densities

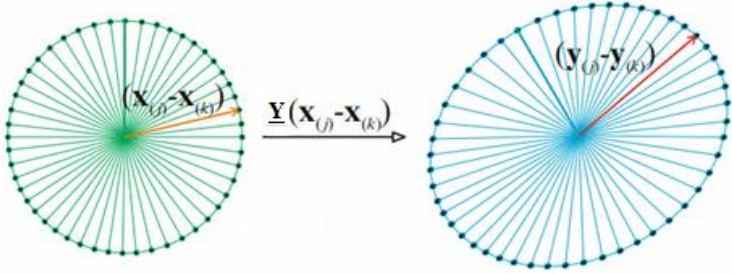

(**b**) Deformation state and force

**Figure 4.** Deformation of vector state and kinematic of PD material point.

As shown in Figure 4b, the motion of the PD material point is controlled by combined action within a finite range called horizon, $H_x$. In the three-dimensional PD framework, a material point's horizon is described as a sphere with a finite radius called the horizon size

denoted by δ. Material particles surrounding the given material particle are called family members. The kinematics of PD material particles are defined as

$$
\begin{cases}
\underline{\mathbf{Y}}\left(\boldsymbol{x}_{(k)}, t\right) = \begin{Bmatrix} \boldsymbol{y}_{(1)} - \boldsymbol{y}_{(k)} \\ \vdots \\ \boldsymbol{y}_{(\infty)} - \boldsymbol{y}_{(k)} \end{Bmatrix} \\
\left(\boldsymbol{y}_{(j)} - \boldsymbol{y}_{(k)}\right) = \underline{\mathbf{Y}}\left(\boldsymbol{x}_{(k)}, t\right)\left\langle \boldsymbol{x}_{(1)} - \boldsymbol{x}_{(k)} \right\rangle = \boldsymbol{F}\left(\boldsymbol{x}_{(j)} - \boldsymbol{x}_{(k)}\right)
\end{cases}
\tag{14}
$$

where $\underline{\mathbf{Y}}\left(\boldsymbol{x}_{(k)}, t\right)$ represents the deformation vector state in the reference configuration and $\left(\boldsymbol{y}_{(j)} - \boldsymbol{y}_{(k)}\right)(j = 1, 2, \cdots, \infty)$ is defined as position vectors within the finite range of the given material particle $\boldsymbol{x}_{(k)}$ The position vectors form an infinite-dimensional array $\underline{\mathbf{Y}}\left(\boldsymbol{x}_{(k)}, t\right)$. Compared with classical theory, the deformation vector state $\underline{\mathbf{Y}}\left(\boldsymbol{x}_{(k)}, t\right)$ and the second-order tensor $\boldsymbol{F}$ can build mathematical relations in the PD framework. This mathematical relation is defined as the expansion of the tensor $\boldsymbol{F}$.

As shown in Figure 5, PD theory describes the interaction process of material points in space using the nonlocal method. Fatigue cracks are naturally included in the space integral motion control equation (SIE). Elastic–plastic fracture mechanics (EPFM) theory characterizes the stress–strain response process of material points using the local method. Fatigue cracks are treated as boundary conditions of structural components or the space partial differential motion control equation (SPDE). Therefore, the PD theory permits crack initiation and growth and multiple cracks are generated in an autonomous fashion under fatigue load.

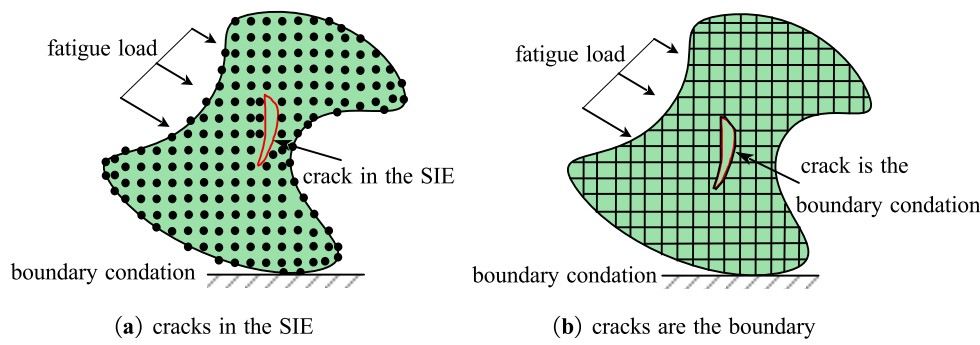

(**a**) cracks in the SIE  (**b**) cracks are the boundary

**Figure 5.** The difference between PD and EPFM.

### 3.2. Energy-Based Failure Model

As shown in Figure 6a, in one loading cycle, the force on the bond varies irregularly as the change in displacement between the two materials is erratic. The bond $\xi$ is subjected to the deformation vector states $\underline{\mathbf{T}}\left[\boldsymbol{x}_{(k)}, t\right]$ and $\mathbf{T}\left[\boldsymbol{x}_{(j)}, t\right]$ acting on both ends of the rod. When a relative displacement occurs between the material particles $\boldsymbol{x}_{(k)}$ and $\boldsymbol{x}_{(j)}$, the work completed on the spring-like bond can be obtained by integrating the deformation vector state. The energy density in the bond $\xi$ during one loading cycle is performed as follows

$$
\psi_{\xi} = \int_{\eta(t_1)}^{\eta(t_2)} \left\{ \underline{\mathbf{T}}\left[\boldsymbol{x}_{(k)}, t\right]\left\langle \boldsymbol{x}_{(j)} - \boldsymbol{x}_{(k)} \right\rangle - \underline{\mathbf{T}}\left[\boldsymbol{x}_{(j)}, t\right]\left\langle \boldsymbol{x}_{(k)} - \boldsymbol{x}_{(j)} \right\rangle \right\} \cdot d\eta
\tag{15}
$$

where $\psi_{\xi}$ is defined as units of energy stored in the bond $\xi$ subjected to deformation vector states acting on both ends of the rod. The path of integration is from some the scalar value of displacement $\eta(t_1)$ to the final value $\eta(t_2)$, which is a function of the time in the whole integration process. Based on the non-ordinary state-based (NOSB) peridynamics theory,

the direction of the deformation vector states $\underline{T}\left[x_{(k)}, t\right]$ and $T\left[x_{(j)}, t\right]$ are not necessary along the bond direction and their values may not equal.

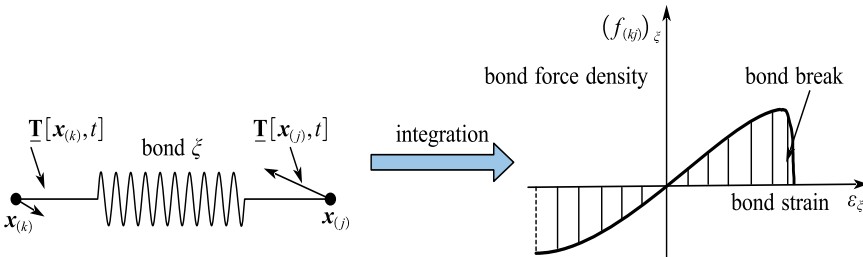

(**a**) The force on peridynamic bond

(**b**) Bond energy in one loading cycle

**Figure 6.** Bond damage and energy density.

It can be seen that the work acting on a solid body is similar to the above integration process, in which the physics equation comes from the integration over the path length. Based on classical continuum medium mechanics, the work performed over the path is performed as

$$W_s = \int F \cdot ds \tag{16}$$

where $W_s$ represents the work performed along the path $s$ and $F$ is the force acting on the structure or the body. The whole process conforms to the theorem of work and kinetic energy.

Compared with Equations (15) and (16), it is clear that the calculation method of work $W_s$ carried out on the body is equivalent to $\psi_\xi$ performed on the bond in peridynamic solid materials, the force $F$ acting on the structure or the body is equivalent to deformation vector states acting on both ends of the rod.

As shown in Figure 7, all points $P_{down}$ along the dashed line $l$, connected to all points $P_{up}$ across fracture plane $\gamma$ of unit area and the volume above the fracture plane. The energy density required to open the plane $\gamma$ is as follows:

$$w_\gamma = \int_0^\delta \int_0^{2\pi} \int_z^\delta \int_0^{\cos^{-1}(z/\xi)} w_c \, \xi^2 \sin\phi \, d\phi \, d\xi \, d\theta \, dz \tag{17}$$

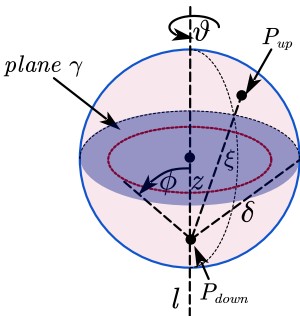

**Figure 7.** Fracture surface in a peridynamic solid.

### 3.3. Fatigue Crack Tip Analysis in the Peridynamic Solid Body

As shown in Figure 8a, the direction of critical bonds is vertical with the crack model I. As the crack propagates along the $x$-direction, the bonds break progressively under cycle loading within the horizon. For a given critical bond, the bond elongation is largest compared with other remaining and nearby bonds. Based on the elastoplastic theory, the amount of critical bond elongation $s^*_{critical}$ can be calculated based on the elastoplastic theory.

$$s^*_{critical}(\delta) = \hat{s}_{critical} \frac{\lambda}{E\sqrt{\delta}} \tag{18}$$

where $\hat{\lambda}$ represents elastic–plastic stress intensity factor in the framework of EPFM, $\hat{s}_{critical}$ is a nondimensional parameter, called elongation coefficient, $\delta$ is the horizon radius, and $E$ is the elastic modulus for elastic–plastic deformation.

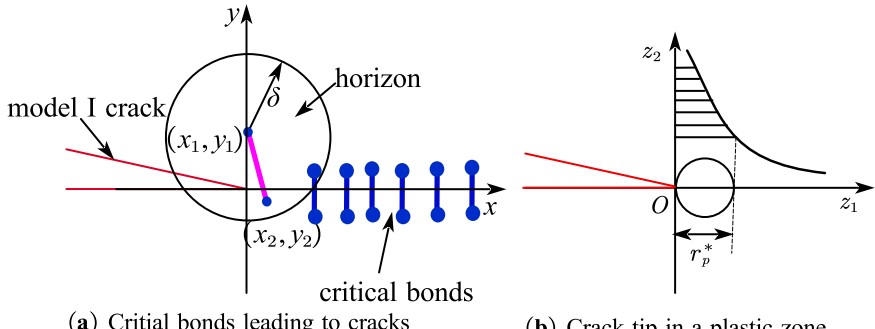

(**a**) Critial bonds leading to cracks   (**b**) Crack tip in a plastic zone

**Figure 8.** Crack tip analysis with model I crack.

As shown in Figure 8b, the critical bond and its neighboring breaking result in a plastic zone near the fatigue crack tip. The radius of the elastic–plastic zone $r_p^*$ is larger than the value $z$, which is in the coordinate systems $\{z_1, z_2\}$. In this elastic–plastic zone, the strain distribution of the peridynamic bonds approaches the strain distribution of finite elements when the relationship $z \gg \delta$ satisfied. Based on the elastoplastic theory, the peridynamic bonds strain in the zone is performed as follows:

$$s(z) = \frac{\hat{\lambda}}{E\sqrt{2\pi z}}\left(0 \leqslant z \leqslant r_p^*\right) \tag{19}$$

Combining Equations (18) and (19), a function $\hat{f}\left(\frac{z}{\delta}\right)$ independent of external cyclic loading is expressed as

$$\hat{f}\left(\frac{z}{\delta}\right) = \frac{1}{\hat{s}_{critical}\sqrt{2\pi z/\delta}} \tag{20}$$

## 4. Fatigue Model Based on Non-Ordinary State-Based Peridynamics

In this section, a non-ordinary state-based peridynamic fatigue model is built based on the mechanism of fatigue. For the crack-nucleation stage, the damage evolution starts from the first bond break, and its surrounding bonds show progressive damage as the loading cycle increases. Each bond in the body is defined as the ideal fatigue test specimen under variable loads. Material points in a peridynamic solid are connected via physical interactions. As the bonds break within the finite range, the obliteration of the physical interactions occurs irreversibly. The spring-like bond between two particles, $k$ and $j$, breaks irreversibly. Therefore, the fatigue loading is redistributed within the peridynamic solid body in each loading cycle, leading to progressive fatigue damage initiation and propagation autonomously.

*4.1. Progressive Failure in the Non-Ordinary State-Based Peridynamic Body*

As shown in Figure 9a, the fictitious bond $\xi_{kj}$ in the peridynamic solid is subjected to the deformation vector states under the tension–tension fatigue loading type. The progressive fatigue damage model exerted on the bond can be performed as follows:

$$D_{kj}\left(\xi_{kj}, t\right) = \begin{cases} 1, & \text{if } \xi_{kj} \text{ is broken} \\ \vartheta\left(\xi_{kj}, s_{kj}, t\right), & \text{otherwise} \end{cases} \tag{21}$$

where $\vartheta\left(\xi_{kj}, s_{kj}, t\right)$ is a normalized bond $\xi_{kj}$ damage function, depending on the current bond deformation $S_{kj}$ and the history time $t$.

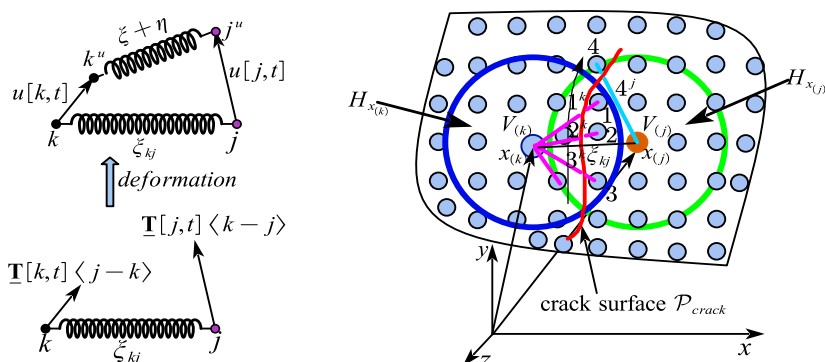

(**a**) particles motion and bond $\xi_{kj}$ defomation  (**b**) progressive failure leading to crack suface

**Figure 9.** Bond deformation and crack surface in PD configuration.

As shown in Figure 9b, for a given material particle $x_{(k)}$, the bonds $1^k$, $2^k$, and $3^k$ within their horizon family $H_{x(k)}$ break in sequence and the physical interactions of the two particles vanish. Meanwhile, for the material particle $x_{(j)}$, the bond $4^k$ within its horizon family $H_{x(j)}$ breaks after the other three bonds break. The four broken bonds show the crack surface $p_{crack}$ (curve line in red), and its direction is illustrated by the point of the arrow.

The point damage is defined as the weighted ratio of the number of eliminated bonds interactions to the total number of initial interactions within its family. Therefore, the fatigue damage of a particle $x_{(k)}$ within its horizon family under tension–tension fatigue loading can be described as

$$D_k\left(x_{(k)}\right) = \frac{\int_{H_{x(k)}} \vartheta\left(\xi_{kj}, s_{kj}, t\right) dV'}{\int_{H_{x(k)}} dV'} \tag{22}$$

where $dV'$ is an incremental volume for the material point connecting the point $x_{(k)}$ within its horizon family $H_{x(k)}$.

### 4.2. Multiscale Fatigue Model for Multi-Crack Process

Metal fatigue failure is a cross-scale damage evolution process involving many physical levels. The introduction of the multiscale method promotes the establishment of a fatigue model based on physical mechanisms. However, there are still some problems in the research of multiscale fatigue, such as the imperfect theory of cross-scale correlation mechanics and the lack of an effective method of cross-scale experiment characterization. The process of fatigue, crack initiation, and slow crack propagation shows a trans-scale characteristic, mainly manifested in time and space. Over time, from milliseconds to years, progressive fatigue damage causes crack initiation and propagation. Over space, fatigue crack length is characterized from micrometer to meters.

As shown in Figure 10b, 3D geometric specimen undergoes tension–tension fatigue loading acting on the boundary layer region. The boundary layer region is discretized into material particles and uniform loads were performed on each discrete material point in the boundary layer region (as shown in Figure 10c). For a given bond $\xi$ in the peridynamic solid, as shown in Figure 11a,b, the deformation state acting on the fictitious bond $\xi$ varies nonlinearly because after every loading cycle to the next fatigue cycle, the fatigue loading is redistributed within the peridynamic solid body, leading to progressive fatigue damage. The fictitious bond strain in the peridynamic solid body varies irregularly during each loading cycle, as shown in Figure 11c.

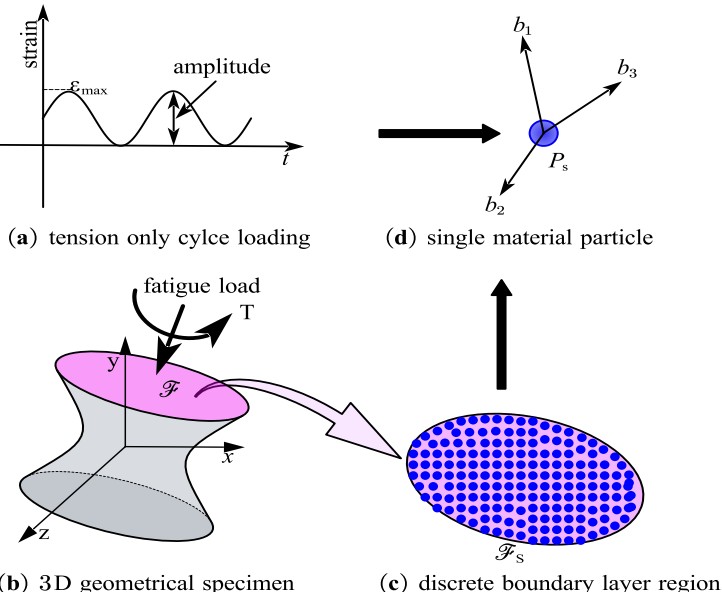

**Figure 10.** Fatigue loading exerted on the boundary layer region.

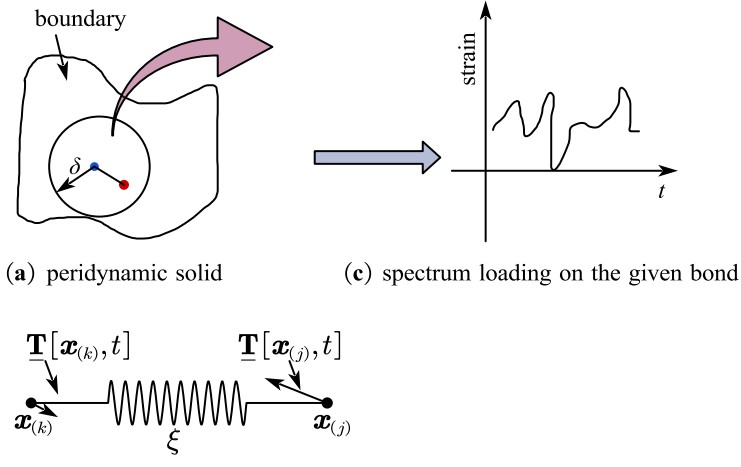

**Figure 11.** Spectrum loading on the material point during loading cycle.

In classical theory, a given element only interacts with other elements in a local range; out of the range, there is no interaction. Therefore, the stress state value of the element can only be calculated through the closest neighbors' deformation. This force interaction is reasonable without considering microstructure evolution on the macroscale. However, from the point of view of atomic theory, the existence of long-range force is obvious. With the decrease in the geometric length scale and close to the atomic scale, the assumption of local force interaction is broken. Therefore, the local force interaction assumption becomes a controversial question across various scales. The fracture mechanics mainly focus on the existing macroscopic cracks rather than the initiation of cracks. Even with existing macroscopic cracks, the validity of the local force interaction assumption becomes ambiguous.

There is a direct mathematical relationship between the current stress and fatigue damage accumulation rate of the given bond $\xi$. The damage evolution rate is determined as follows:

$$\frac{dD}{dN} = C \exp(-\frac{Q_c(\sigma_a, \sigma_m, D)}{kT})$$

(23)

where $Q(\sigma_a, \sigma_m, D)$ is the critical active energy related to the mean stress and stress amplitude and $D$ is the damage quantity during one loading cycle. Based on the widespread

fatigue damage and microscopic fatigue crack initiation and propagation law, the critical active energy is confirmed with inverse analysis.

Combining Equations (12) and (23), the normalized progressive damage quantity of the given bond $\xi$ is determined using fictitious time $t_f$ as follows:

$$\begin{cases} \lambda\left(\boldsymbol{x}, \xi, t_f\right) = N[1 - (1-D)^{B+1}]^{A(\widetilde{\sigma})}(\dfrac{\sigma_a}{M(\overline{\sigma})(1-D)})^{B} \\ t_f = \dfrac{N}{N_f} \end{cases} \tag{24}$$

where $N_f$ is the failure cycle number; $N$ is the present cycle number, the smallest cycle number to failure; $N$ is the current cycle number; $M(\overline{\sigma})$ is the function of average stress; $A(\widetilde{\sigma})$ is the function of alternating stress; and $B$ is the material characteristic coefficient related to temperature. The uniform damage quantity $\lambda\left(\boldsymbol{x}, \xi, t_f\right)$ means that the bond has a historical damage evolution in the past. The given bond $\xi$ breaks at present cycle number $N$:

$$\lambda(N) \geq 1 \tag{25}$$

The first bond in the peridynamic solid body breaks within the horizon radius. Then, the critical control equation can be used as follows:

$$N_1[1 - (1-D_c)^{B+1}]^{A(\widetilde{\sigma})}(\dfrac{\sigma_a}{M(\overline{\sigma})(1-D_c)})^{B} = 1 \tag{26}$$

where $N_1$ is the cycle number for the given bond breaks with permanent change, and $\lambda(N_1) = 1$. When the cycle number increases, the first bond breaks during the crack-nucleation stage. The current cycle number becomes larger than the current number $N_1$.

$$N_1 \geq \dfrac{1}{C_1[1 - (1-D_c)^{B+1}]^{A(\widetilde{\sigma})}(\dfrac{\sigma_a}{M(\overline{\sigma})(1-D_c)})^{B}} \tag{27}$$

### 4.3. Criterion of Fatigue Multi-Crack Initiation to Propagation

As shown in Figure 12a, the mechanism of multi-crack can be depicted as two continuous stages, crack 1 and crack 2. Each crack includes nucleation and growth stages. When crack 1 is complete, crack 2 begins to nucleate and propagate in an autonomous fashion. As for crack 1, during its nucleation stage, each bond strain is independent of the current cycle number. A bond strain is changing over time as the bond transits to the fatigue crack growth phase. When crack 1 reaches the propagation stage, the bond break numbers should be defined. For the given material particle $x_{(j)}$, within the range of particle $x_{(k)}$, the bond $\xi_{jk}$ (denoted as sky blue) reaches the propagation stage as the particle damage $D_j(x_j)$ satisfies the following:

$$D_j(x_j) \geq 0.5 \tag{28}$$

where $D_j(x_j)$ is defined by Equation (22). At this time, the bond $\xi_{jk}$ transits to the fatigue crack growth phase.

As shown in Figure 12b, a bond near an approaching fatigue crack undergoes cyclic strain, which changes over time, eventually causing the bond to break. The crack growth rate is shown in Equation (20).

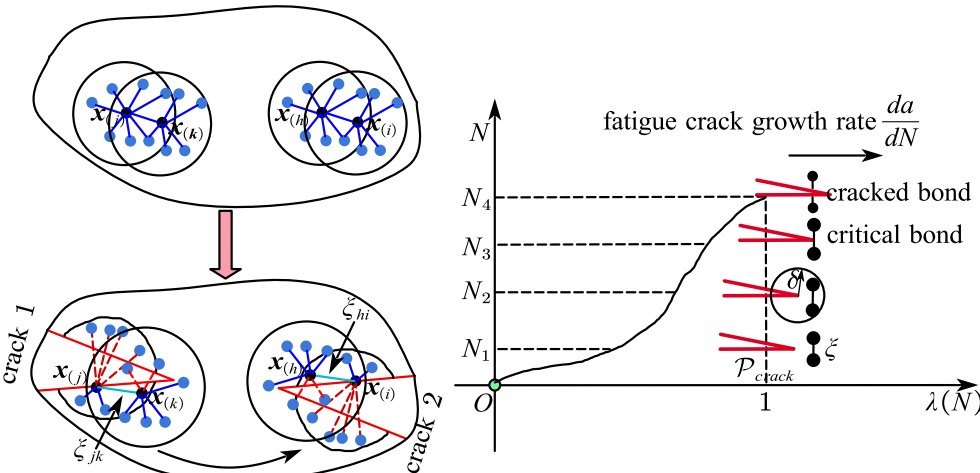

(**a**) multiple crack initiation and propagation (**b**) the bond breaks in a progressive way

**Figure 12.** The crack tip analysis of bond damage.

## 5. Validation Procedure of Numerical Solution Method

### 5.1. Quasi-Static Solution for Each Cyclic Loading

Damage in a peridynamic solid body is accumulated because of cyclic continuous loads on the load point. The quasi-static or static solution simulates cyclic damage in one loading cycle. Therefore, the accumulated damage is determined by solving static problems in every loading cycle. Meanwhile, for the multiple cracks, the control equation is the integral differential during the tension–tension loading pattern.

A new static solution is deduced at the current cycle when some bonds break during one loading cycle. The static solution is obtained with this modified ADR (adaptive dynamic relaxation) method. For a given particle, the motion control equation is integral differential with fictitious inertia terms. As for all particles in a structure, a set of equations is expressed as

$$D\ddot{U}(X,t) + \zeta D\dot{U}(X,t) = F(U,U',X,X') \tag{29}$$

where $D$ is the virtual diagonal density matrix, $\zeta$ is the bulk drag coefficient, $X$ is the original site vector, and $U$ is the initial movement vector. For material particles in the peridynamic solid body, $X$ and $U$ are used as follows:

$$\begin{cases} X^T = \{x_1, x_2, \cdots, x_N\} \\ U^T = \{u(x_1,t), u(x_2,t), \cdots, u(x_N,t)\} \end{cases} \tag{30}$$

where $N$ is the number of all the material points in the configuration body. Finally, the vector $F$ is composed of PD interaction and body forces and its $k^{th}$ component.

$$F_{(k)} = \sum_{j=1}^{M} \left( t_{(k)(j)} - t_{(j)(k)} \right) \left( v_{cj} V_{(j)} \right) + b_{(k)} \tag{31}$$

where $M$ is all the material particles in the finite range of the given particle $x_{(k)}$ and $v_{cj}$ is the volume modification coefficient for the given material particle $x_{(j)}$. The next iteration step of velocities and displacements can be described as

$$\begin{cases} V^{n+1/2} = \dfrac{\left( (2 - \zeta^n \Delta t) V^{n-1/2} + 2\Delta t D^{-1} F^n \right)}{2 + \zeta^n \Delta t} \\ U^{n+1} = U^n + V^{n+1/2} \Delta t \end{cases} \tag{32}$$

where $n$ is the number of iteration steps. The elements on the diagonal line of the density matrix, $D$, are as follows:

$$\gamma_{kk} \geqslant \frac{1}{4}\Delta t^2 \sum_j \left| \mathcal{J}_{kj} \right| \tag{33}$$

where $\mathcal{J}_{kj}$ is the rigidity matrix of the peridynamic material solid under the small placement hypothesis. Additionally, the rigidity matrix is expressed as follows.

$$\sum_j \left| \mathcal{J}_{kj} \right| = \sum_{j=1}^{M} \frac{\left| \xi_{(k)(j)} \cdot e \right|}{\left| \xi_{(k)(j)} \right|} \frac{4\delta}{\left| \xi_{(k)(j)} \right|} \left\{ \frac{ad^2\delta}{\left| \xi_{(k)(j)} \right|} \left( v_{ck}V_k + v_{cj}V_j \right) + b \right\} \tag{34}$$

where $e$ is a unit vector along the non-diagonal direction and $a$, $b$, and $d$ are regulating constants. The elements of the rigidity matrix can be determined by the summation given in Equation (31).

*5.2. Equivalent Fatigue Parameters between PD and EPFM*

Multiple crack expansion in each iteration is controlled by the boundary equation, which leads the multiple cracks to propagate in the material configuration. Naturally, this boundary equation can be expressed by the stress intensity factor (SIF). It is difficult to compute three-dimensional SIF in the framework of EPFM because of its complexity in mathematics and physics, and singularity appears during the calculation process. Based on the non-ordinary state-based PD theory, the driving force is calculated with the bond damage model.

As shown in Figure 13, to build a correlation between the multiple crack tip in PD and EPFM, an equivalent stress intensity factor $\lambdabar_{equ}$ along the crack front is described as

$$\lambdabar_{equ} = \int_\Gamma \left( Wdy - T_i \frac{\partial u_i}{\partial x}ds \right) = \frac{2E}{(1-v^2)} \sum_{i=I,II,II} \int_i (W_idy - w_ids) \tag{35}$$

where $x = x_1$, $y = x_2$, and $z = x_3$ form the 3D rectangular coordinate system; the origin coordinate system is at the multiple-fatigue crack tip; $\lambdabar_{equ}$ is calculated through the integral around a contour with the multiple crack surface; $T_i = \sigma_{ij}n_j$ is a pulling traction arrow along the multiple crack surface; $\sigma_{ij}$ is a stress unit normal vector; $n_j$ is the outward unit normal vector; $u_i$ is a movement vector; and $ds$ is an element of the crack surface. By using the surface integration, the relationship between the displacement and fatigue crack is built. Each iteration step for multiple crack propagation can be obtained from the above mathematical model.

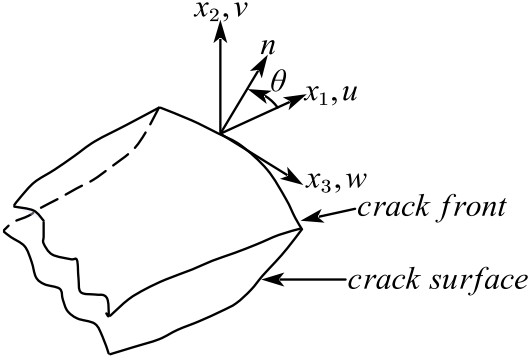

**Figure 13.** Equivalent fatigue parameters between PD and EPFM.

### 5.3. The Whole Process of Simulating Fatigue Multi-Fatigue Crack Initiation and Propagation

Similar to the classical process with the framework of EPFM, the process of multiple-fatigue-crack nucleation and propagation can be depicted with the framework of non-ordinary state-based PD theory, as shown in Figure 14.

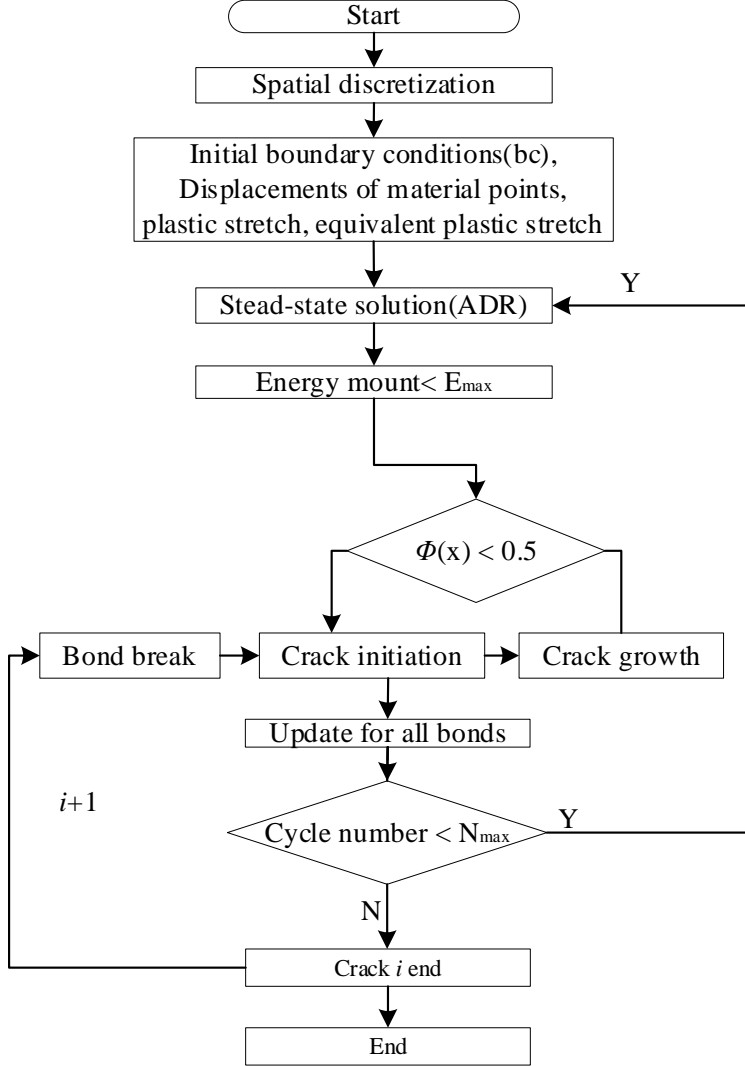

**Figure 14.** Flowchart of the multiple-crack simulation.

## 6. Fatigue Experiment

### 6.1. Structural Fatigue Testing Platform with Overalls Tools

It can be seen in Figure 15 that the fatigue test experiment was conducted in a structural test platform with overalls tools. The servo-hydraulic testing system provides a combined load with axial and torsion on the aircraft wing corner box. The servo-hydraulic dynamic test platform has an axial load capacity of $\pm100$ KN ($\pm22.4$ kip) and a torque capacity of $\pm1000$ Nm ($\pm8850$ in-lb), with an actuator stroke of 50 mm along the axial direction and a rotary stroke of $45°$ along the circumference.

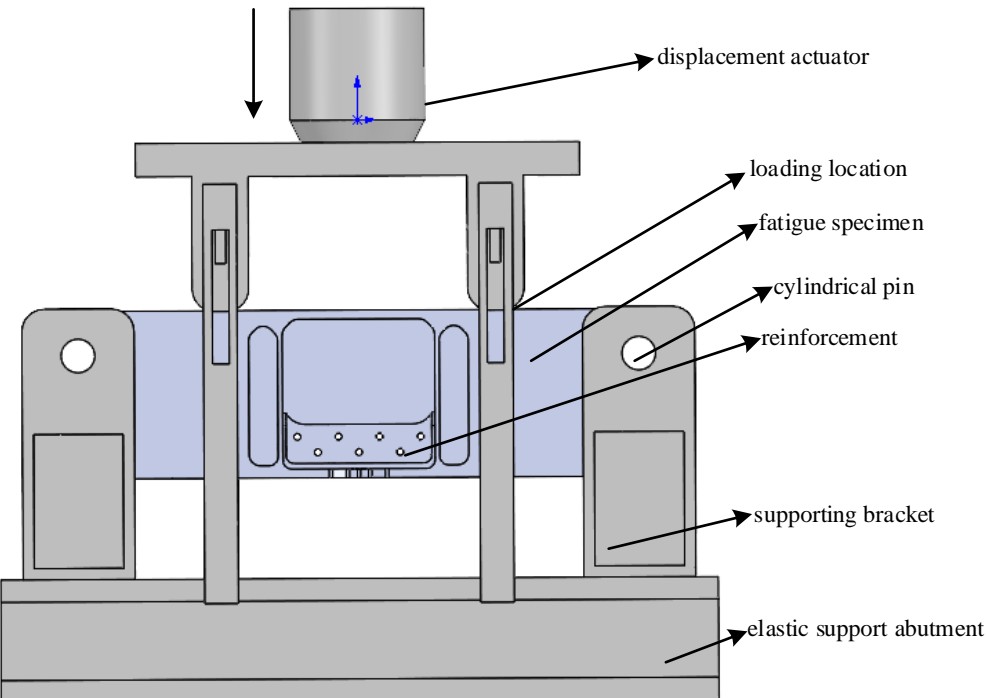

**Figure 15.** Fatigue test platform for the tension-only experiment.

As shown in Figure 15, the displacement actuator provides tensile or compressive loads which is confirmed by the user. Based on the loading state for the engineering structure tested on the experimental platform, the zero point of piston is confirmed firstly and then the displacement actuator moves downward. When the load reaches some value, the actuator returns to move upward but will not cross the zero point. As for the fatigue specimen, it can be seen that the tension–tension loading of the model I crack and the direction of fatigue crack propagation is vertical to the tension load on the crack front. The load ratio R is defined as the ratio of the minimum and maximum loads. Therefore, the stress ratio R is chosen to be close to but not exactly zero such that R = 0.1. Thus, $\sigma_{min} = R \times \sigma_{max}$, where $\sigma_{max}$ is the desired maximum stress. Without environmental effects, the load ratio has a more significant effect on the stages I and III fatigue crack growth rates than in stage II. The fatigue test characteristics are 6 Hz frequency and 0.1 as the minimum to maximum loading ratio in the tensile regime.

As shown in Figure 16, the dynamic fatigue testing system with the fatigue test procedure issues control instructions to implement the test process using a central control computer, including waveforms generated by function generators in both axes, signal calibration strategy and related impact factors, a limit set up by human rationality and the system range, and status monitoring via the master system's display interface module.

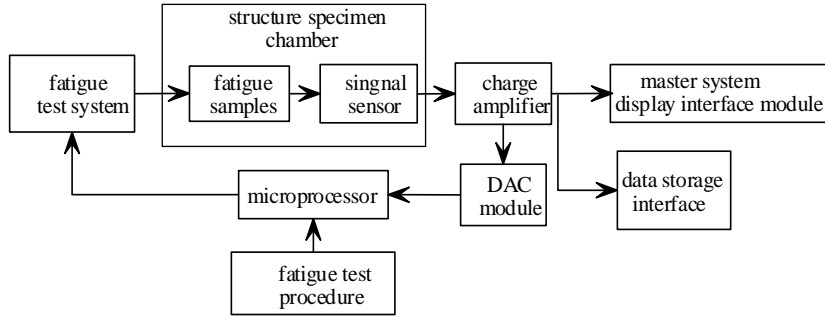

**Figure 16.** Configuration of the structure fatigue test platform.

### 6.2. Sample Dimension Description and Input Parameters

As shown in Figure 17, a computer data recording system and crack length measurement are used for multiple-crack nucleation and expansion. The aircraft wing corner box and the reinforcement are bolted together. A u-shaped notch is present in the middle of the aircraft wing corner box, which also means that the notch is the weakest position of the sample regardless of other environmental factors under tension–tension fatigue loading. The relative percentage contents of elements in the aluminum alloy material are shown in Table 1. The main mechanical property parameters of the aluminum alloy material at room temperature are shown in Table 2. Fatigue loading parameters under tension–tension loads are shown in Table 3.

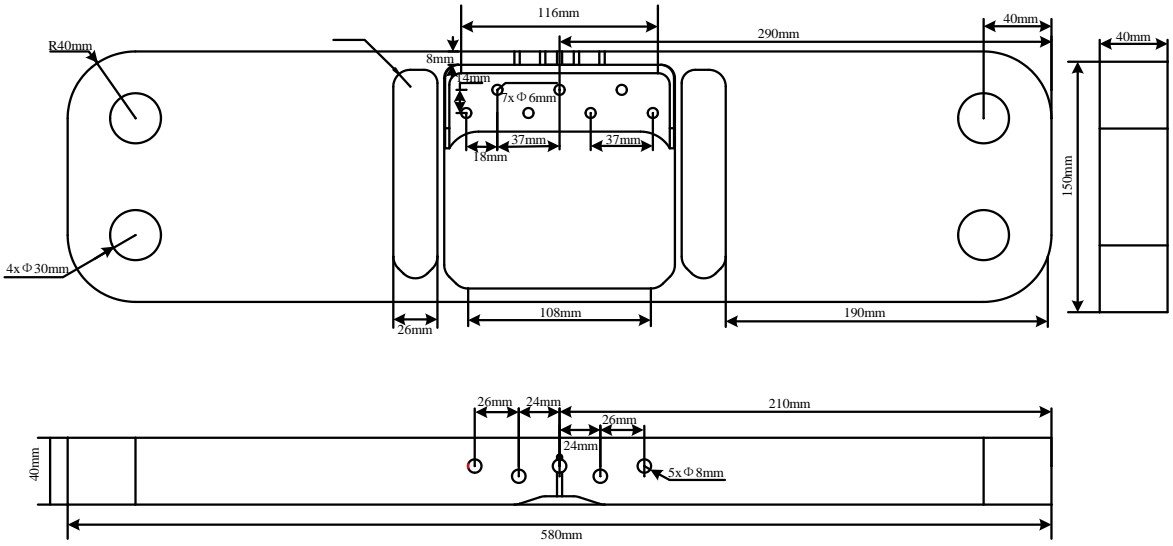

**Figure 17.** Fatigue structure sample.

**Table 1.** The relative percentage contents of elements in aluminum alloy material.

| Zn% | Cu% | Mg% | Mn% | Fe% | Si% | Cr% | Ti% | Al% |
|-----|-----|-----|-----|-----|-----|-----|-----|-----|
| 5.75 | 1.57 | 2.49 | 0.29 | 0.28 | 0.27 | 0.18 | <0.1 | Others |

**Table 2.** Main mechanical property parameters of aluminum alloy material at room temperature.

| Name | E/GPa | $\sigma_{0.2}$/MPa | $\sigma_b$/MPa | $\delta$% | Fatigue Strength/MPa | Density g/cm$^3$ | Poisson' Ratio | Brinell Hardness/HB | Shear Modulus/GPa | Shear Strength/MPa |
|------|-------|---------|---------|-----|----------------------|------------------|----------------|---------------------|-------------------|--------------------|
| 7075-T6 | 70 | 480 | 560 | 7.9 | 160 | 3.0 | 0.32 | 150 | 26 | 330 |

**Table 3.** Fatigue loading parameters under tension–tension loads.

| $\sigma_a$/MPa | $\tau_a$/MPa | $\sigma_{equ}$/MPa | $R_a$ | $F_{max}$/KN | $f$/HZ | Stress Ratio R |
|----------------|--------------|--------------------|-------|--------------|--------|----------------|
| 340 | 196.3 | 480.83 | $\sqrt{3}$ | 9.6 | 6 | 0.1 |

### 6.3. Crack Morphology and Data Recording

The multi-crack morphology of the fatigue sample is shown in Figure 18.

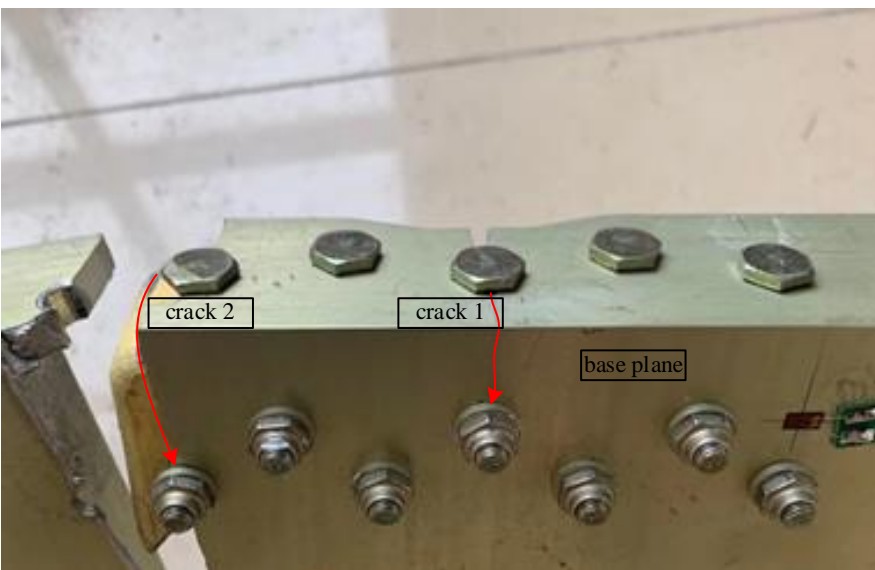

**Figure 18.** Macro multi-crack characteristics of the fatigue sample.

The number of cycles for crack 1 and crack 2 is shown Table 4.

**Table 4.** Cycle numbers recorded for two associated cracks.

| Samples | Crack 1 | | Crack 2 | |
|---|---|---|---|---|
| NO | Initiation | Propagation | Initiation | Propagation |
| 7075-T6-1 | 24,146 | 23,854 | 28,036 | 18,558 |
| 7075-T6-2 | 22,334 | 24,706 | 27,586 | 15,414 |
| 7075-T6-3 | 30,036 | 18,234 | 29,766 | 12,048 |
| 7075-T6-4 | 29,058 | 19,028 | 35,696 | 10,068 |
| 7075-T6-5 | 20,048 | 25,703 | 34,330 | 10,268 |
| 7075-T6-6 | 24,100 | 30,854 | 25,004 | 18,500 |

*6.4. Data Analysis and Countermeasures*

As shown in Figure 18, using the macroscopic crack observation technique and data recording technology, we can see that the aircraft wing corner box has multiple cracks, crack 1 and crack 2. The cycle numbers of crack 1 and crack 2 are recorded in Table 4.

As shown in Table 4, for cycle numbers during initiation, the average fatigue cycle numbers are 30,069 for the fatigue crack 1 life; the average fatigue cycle numbers are 22,106 for the fatigue crack 2 life. When summing these two life periods, the entire life of the multiple cracks is 52,175.

When comparing Figure 19's illustration of fatigue crack 1's path to Figure 18, we can see that the angle between crack 1 and the base plane is about 89.5°. When comparing Figure 20's crack 2 path to Figure 18, we can see that the angle between crack 2 and the base plane is about 87.6°. This simulation result correlates well with the average test result, as shown in Figure 18.

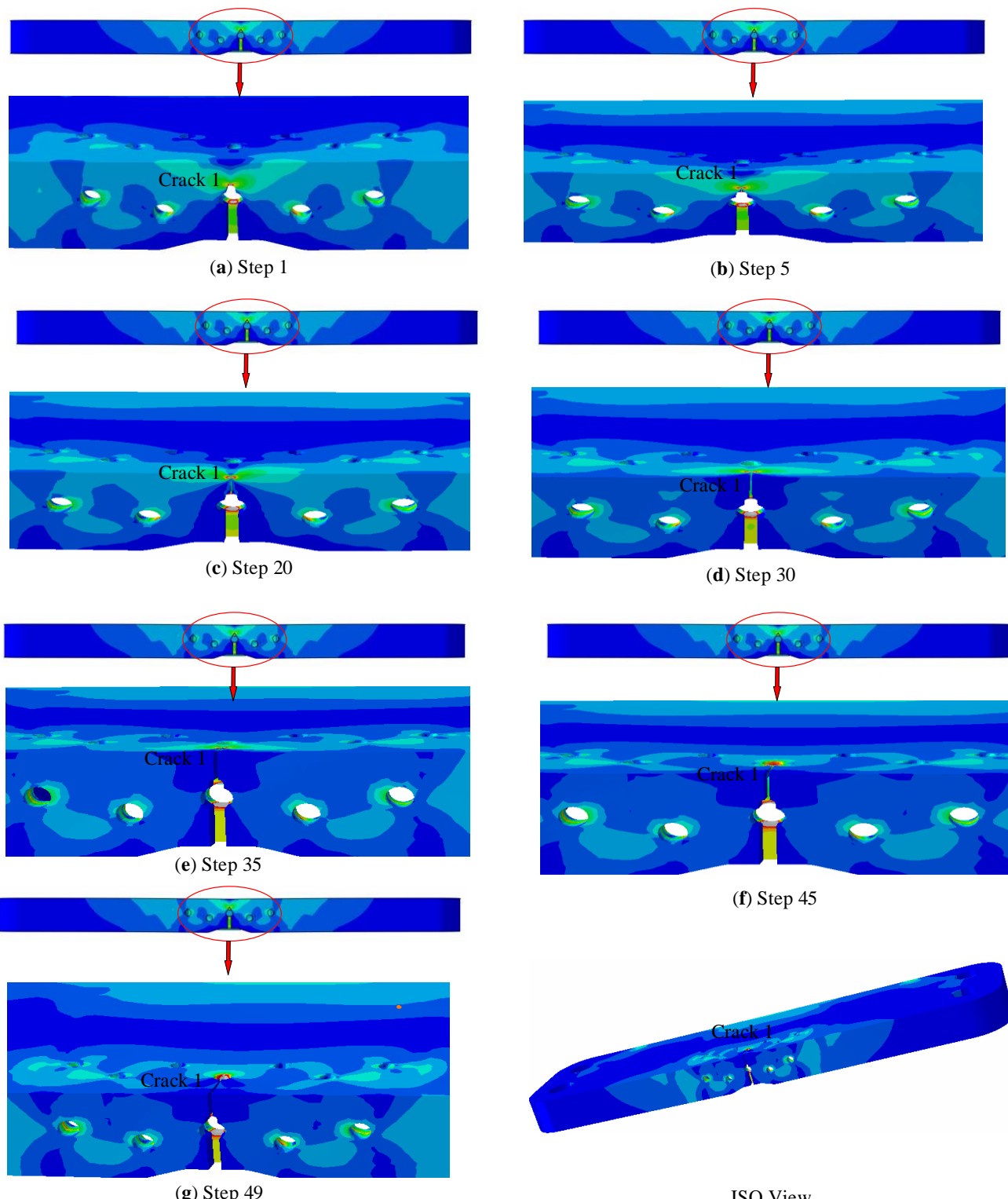

**Figure 19.** Simulation results of crack 1 with framework of non-ordinary peridynamic theory. (**a**) Crack 1 propagates under iterative step = 1; (**b**) Crack 1 propagates under iterative step = 15; (**c**) Crack 1 propagates under iterative step = 20; (**d**) Crack 1 propagates under iterative step = 30; (**e**) Crack 1 propagates under iterative step = 35; (**f**) Crack 1 propagates under iterative step = 45; (**g**) Crack 1 propagates under iterative step = 49.

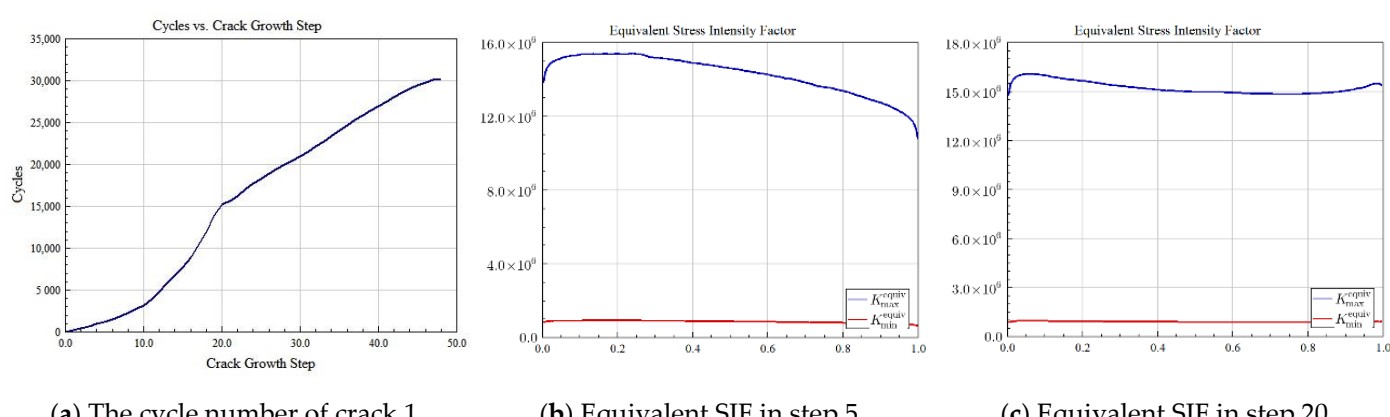

**Figure 20.** Simulation results of the crack 1 and crack 2 with framework of non-ordinary peridynamic theory. (**a**) Crack 2 propagates under iterative step = 1; (**b**) Crack 2 propagates under iterative step = 10; (**c**) Crack 2 propagates under iterative step = 15; (**d**) Crack 2 propagates under iterative step =19.

As shown in Figures 21 and 22, the value of the equivalent stress intensity factor $\lambda_{equ}$ becomes larger as the number of iterations increases.

(**a**) The cycle number of crack 1.      (**b**) Equivalent SIF in step 5.      (**c**) Equivalent SIF in step 20.

**Figure 21.** *Cont.*

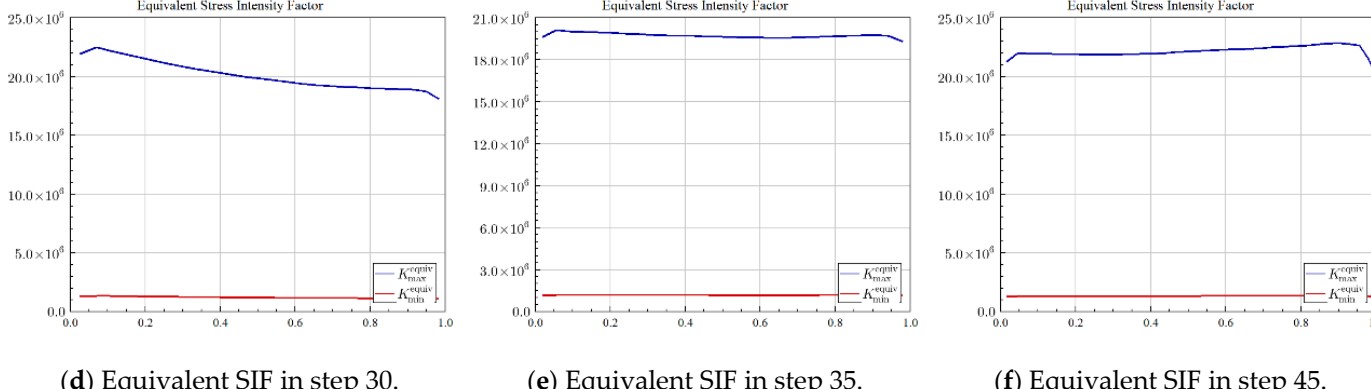

(**d**) Equivalent SIF in step 30.　　　(**e**) Equivalent SIF in step 35.　　　(**f**) Equivalent SIF in step 45.

**Figure 21.** In each specified iteration, the equivalent intensity factors along crack 1 front (**a**) cycle number vs. crack growth step; (**b**) $\lambdabar_{equ}$ along crack propagation front with iterative step = 5; (**c**) $\lambdabar_{equ}$ along crack propagation front with iterative step = 20; (**d**) $\lambdabar_{equ}$ along crack propagation front with iterative step = 30; (**e**) $\lambdabar_{equ}$ along crack propagation front with iterative step = 35; (**f**) $\lambdabar_{equ}$ along crack propagation front with iterative step = 45.

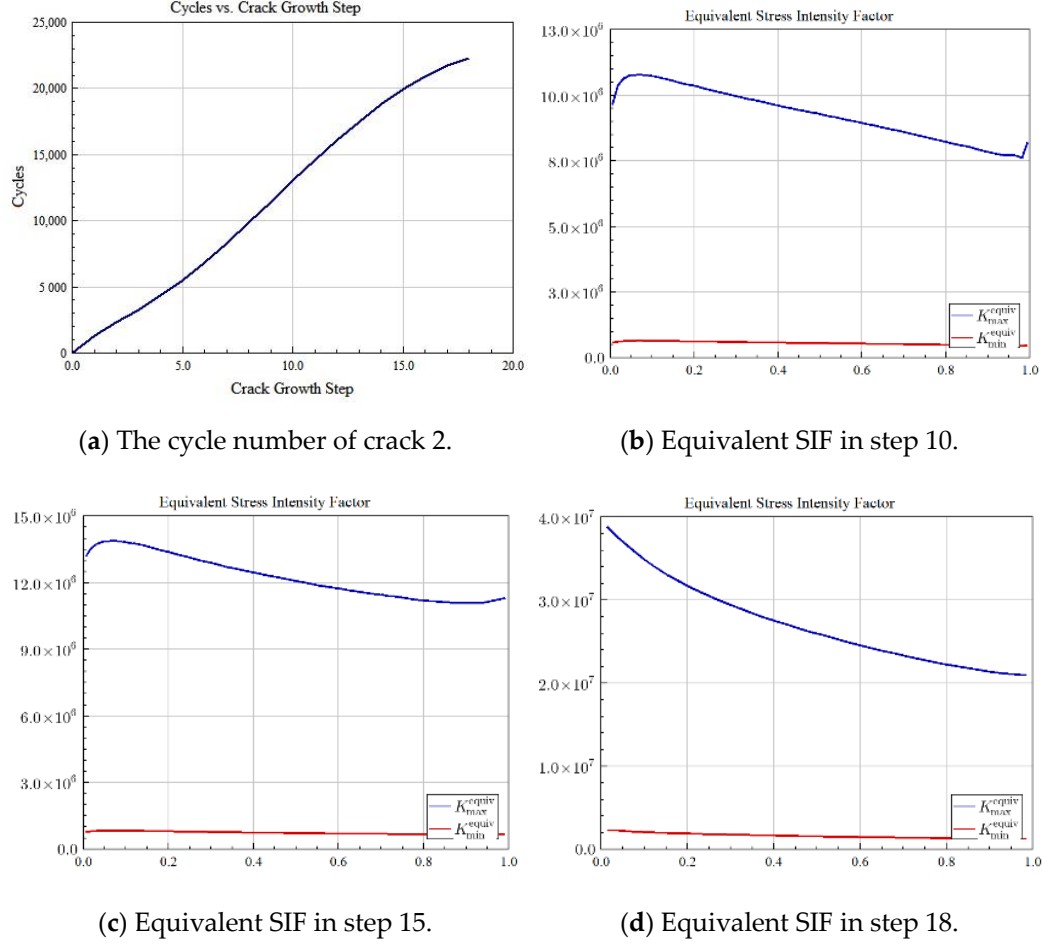

(**a**) The cycle number of crack 2.　　　　(**b**) Equivalent SIF in step 10.

(**c**) Equivalent SIF in step 15.　　　　(**d**) Equivalent SIF in step 18.

**Figure 22.** In each specified iteration, the equivalent intensity factors along crack 2 front. (**a**) Cycle number vs. crack growth step; (**b**) $\lambdabar_{equ}$ along crack propagation front with iterative step = 10; (**c**) $\lambdabar_{equ}$ along crack propagation front with iterative step = 15; (**d**) $\lambdabar_{equ}$ along crack propagation front with iterative step = 18.

## 7. Conclusions

(1) To evaluate fatigue crack life of the aircraft wing corner box under tension–tension fatigue loading, a novel NOSBPD fatigue model for multiple-crack nucleation and propagation was deduced. The multiple cracks nucleate and propagate autonomously with this constitutive model for fatigue.

(2) The proposed non-ordinary state-based peridynamics damage model shows no scale constraints and the whole multiple crack propagation process can be applied to the model. Therefore, the NOSBPD fatigue model successfully addressed cross-scale issues during the multiple-crack lifetime.

(3) Two of the cracks, crack 1 and crack 2, converged and grew into principal cracks according to the time sequence. The numerical calculation results from the proposed model agree well with the experimental results. It is more accurate and effective at reproducing the multiple-crack characteristics, such as spatial warping and multiple original positions, than classical fatigue models according to our comparison.

(4) The natural nucleation and propagation of multiple cracks are obtained with no extra rules to guide the crack propagation. Quantitative analysis of the fatigue damage is obtained. The assessment of three-dimensional multi-crack nucleation for fatigue life prediction is confirmed with the NOSBPD model.

**Author Contributions:** Conceptualization, J.H. and G.W.; methodology, G.W. and R.C.; software, X.Z. and R.C.; validation, J.H., G.W. and X.Z.; formal analysis, J.H. and G.W.; investigation, J.H. and X.Z.; resources, J.H. and R.C.; data curation, X.Z. and G.W.; writing—original draft preparation, J.H. and R.C.; writing—review and editing, J.H and R.C.; visualization, J.H. and R.C.; supervision, W.C.; project administration, W.C.; funding acquisition, W.C. All authors have read and agreed to the published version of the manuscript.

**Funding:** This research was funded by TEN THOUSAND TALENTS PROGRAM, grant number 2018R008.

**Institutional Review Board Statement:** Not applicable.

**Informed Consent Statement:** Not applicable.

**Data Availability Statement:** The data sets generated and/or analyzed during the current study are available from the corresponding author on reasonable request.

**Conflicts of Interest:** The authors declare no conflict of interest.

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
