# Peer review of "Modeling of Multiple Fatigue Cracks for the Aircraft Wing Corner Box Based on Non-ordinary State-based Peridynamics"

_metals, doi:10.3390/met12081286_

Round 1

Reviewer 1 Report

The draft called " Modeling of Multiple Fatigue Cracks for Aircraft Wing Corner Box Based on Non-ordinary Sate Peridynamics " has been significantly improved and to my point of view can be considered as a valuable contribution to Metals.
All the best.

Author Response

Point 1: The draft called " Modeling of Multiple Fatigue Cracks for Aircraft Wing Corner Box Based on Non-ordinary Sate Peridynamics " has been significantly improved and to my point of view can be considered as a valuable contribution to Metals.

All the best.

Response 1: Thank you for your careful review of our manuscript. Thank you again for your approval of our manuscript. The manuscript has been checked again for correct use of grammar and common technical terms. In addition, we have asked again several colleagues who are skilled authors of English language papers to check the English. All minor spell check changes made to the text are in red color. Please check them in the revision mode.

Reviewer 2 Report

The authors have followed my comments and suggestion in a satisfatory way.

Author Response

Point 1: The authors have followed my comments and suggestion in a satisfactory way.

Response 1: Thank you for your careful review of our manuscript. Thank you again for your approval of our manuscript. The manuscript has been checked again for correct use of grammar and common technical terms. In addition, we have asked again several colleagues who are skilled authors of English language papers to check the English. All minor spell check changes made to the text are in red color. Please check them in the revision mode.

Reviewer 3 Report

generally the paer is very intersting but I have fined same mistaks:

l.4 - it is written name zhao. It should be Zhao

l.242 - it is written "vom Mises". It must be written "Huber-Mises"

Tabel 3 - pease define R. Generaly it is possoble to add NOMENCLATUREs

Figures 19 and 22 are too small

I have problem with term "tension-tension" in same parts of paper. If it is corected that limitation for all paper is for R>0. But in this paper I have not fined sach limitation. If yes pease add same comments. If not it shpuld be written more generaly "tension-compresion"

Reviewer 4 Report

The manuscript written with titled as “Modeling of Multiple Fatigue Cracks for the Aircraft Wing 2 Corner Box Based on Non-ordinary State-based Peridynamics” is a good manuscript considering lots ofsimulation work. But the present condition of manuscript warrants some modifications, which are as follows:

Abstract is lacking in quantitative information related to finding of present manuscript.

At some places, grammatical errors are these. Kindly remove them by thorough reading the manuscript.

Rest manuscript is well written and can be recommended for publication

Author Response

This manuscript is a resubmission of an earlier submission. The following is a list of the peer review reports and author responses from that submission.

Round 1

Reviewer 1 Report

The manuscript is a valuable contribution to the area, a topic which occupies and will occupy generations of researchers.

It is internationally agreed that fatigue cracks are considered as a great challenge to understand the involved processes. But, the intensive use of aircraft lead to these fatigue processes and may give rise to questions. It has caught tremendous attention and interest for aircraft safety. All this is because of the unique properties that are at hand, offering possibilities of multifunctionality and applications relating technology. We appreciate particularly your improvement on theorical approach with modelling to study multiple fatigue cracks for aircraft wing corner box.

Recent works on fatigue cracks introduced in the chemical formulation showcase potential risks of particle releases. For example, many studies highlight particle exposures due to material solicitations. Cases of particle exposure in the field of occupational hygiene at workplaces have been reported [1,2]. These aspects are addressed according to metrological challenges related to particle characterization. Furthermore, applications on tribological framework are available [3].

The overall impression of the manuscript is good. It is well-structured and easily readable. However, a last grammar and vocabulary check by a native English speaker would be appropriate.

Secondly, references are missing. It is particularly obvious on the risk based on materials fatigues. You could cut-and-paste related references below (e.g. reference [4] could be very useful for you). Moreover, some references from this journal can be added.

I will now share my observations in order of appearance in the manuscript.

Title: The title is well chosen and indicates clearly the scope and topic of your contribution. However,

Graphical abstract “A Graphical Abstract is a single, concise, pictorial and visual summary of the main findings of the article. This could either be the concluding figure from the article or a figure that is specially designed for the purpose, which captures the content of the article for readers at a single glance. Please see examples below.” I this perspective, your graphical abstract could be improved and become more ‘catchy’ and self-explaining.

Highlights: Research highlights ‘are a short collection of bullet points that convey the core findings and provide readers with a quick textual overview of the article.’ The proposed highlights are a good choice in this view, however you might want to check whether these are ‘stand-alone’ choices, i.e. if the reader could understand the scopes and respective findings. I think there is room for improvement, even if the number limitation of characters is obviously a constraint.

Abstract: ‘An abstract is a brief summary of a research article, thesis, review, conference proceeding, or any in-depth analysis of a particular subject and is often used to help the reader quickly ascertain the paper's purpose.’ In this view, your abstract could be improved, similarly to the previous point,

- By more clearly indicating the purpose of your research. The need for research in this field is certainly a strength of your paper, so tell people about.

- You might give a hint to the quality of your results, i.e. relative improvement, repeatabilities etc. The abstract remains vague with respect to this.

Introduction: ‘The introduction leads the reader from a general research issue or problem to your specific area of research. It puts your research question in context by explaining the significance of the research being conducted. This is usually done by summarizing current understanding (research

to date) and background information about the topic. This is followed by a statement of the purpose of your research issue or problem. This is sometimes followed by a hypothesis or a set of questions you attempt to answer in your research. You may also explain your methodology (how you will research this issue) and explain what your study can reveal. It also may contain a summary of the structure of the rest of the paper.’ The introduction as a whole is corresponding to these needs and very well written. With respect to what has been said before, a minor point would be to introduce the different exposure scenarios in the framework of risk analysis, which are

- Environmental release (you mentioned this and provided a major reference, you might also cite [5])

- Release to the worker (occupational risk, e.g. [4])

Methods / Experimental: Generally, the Methods chapter should allow for getting all the basic details so that the experiments can be reproduced, mentioning all the appropriate controls, including appropriate citations included and the reference of each single material used. I think this chapter in this manuscript fulfills this purpose, but, however, lacks of sufficient basic information of this type: Please add dimensions, parameter ranges etc. In view of its reproducibility, the efforts allowing for improving this might be indicated

Results and discussion: ‘The purpose of a Results section is to present the key results of your research.’ The manuscript fulfills this in a good way.

Conclusions: The conclusions chapter reminds well the reader of the strengths of your central points and summarizes the evidence supporting these.

References suggested (MDPI style devoted to Metal)

1. Jiménez, A.S.; Puelles, R.; Perez-Fernandez, M.; Goméz, P.; Barruetabeña, L.; Jacobsen, N.R.; Suarez-Merino, B.; Micheletti, C.; Manier, N.; Trouiller, B., et al. Safe by design implementation in the nanotechnology industry. NanoImpact 2020, 100267.

2. Shandilya, N.; Marcoulaki, E.; Vercauteren, S.; Witters, H.; Salazar-Sandoval, E.J.; Viitanen, A.K.; Bressot, C.; Fransman, W. Blueprint for the development and sustainability of national nanosafety centers. Nanoethics 2020, 14, 169-183.

3. Philippe, F.; Morgeneyer, M.; Xiang, M.Q.; Manokaran, M.; Berthelot, B.; Chen, Y.M.; Charles, P.; Guingand, F.; Bressot, C. Representativeness of airborne brake wear emission for the automotive industry: A review. P I Mech Eng D-J Aut 2021, 235, 2651-2666.

4. Bressot, C.; Shandilya, N.; Nogueira, E.; Cavaco-Paulo, A.; Morgeneyer, M.; Le Bihan, O.; Aguerre-Chariol, O. Exposure assessment based recommendations to improve nanosafety at nanoliposome production sites. Journal of Nanomaterials 2015, 10 pages.

5. Warheit, D.B. Hazard and risk assessment strategies for nanoparticle exposures: How far have we come in the past 10 years? F1000Res 2018, 7, 376.

Reviewer 2 Report

The authors say to introduce a novel state-based PD fatigue model into the calculation of compression-compression loads. Moreover, in each loading cycle, they redistribut the fatigue loading among the peridynamic solid body, leading to the progressive fatigue damage formation and propagation.

It is very difficult to judge this paper that is not shown as a review paper, but it has some characteristics of the review paper. In my opinion, the authors should decide in one sense or the other, and, consequently work in this sense. Moreover, they should consider the following points.

1. In many cases the explanation of some concepts is transferred  to the figues, but, unfortunately, they are not so clear (i.e. Fig.1,7,8,11)

2. Many equations are not well explained. It seems that they are simply reported by other papers (i.e. Eq.21,30,31).

Thus, even if the topic can be considered interesting, the paper cannot be accepted in this form. 

Reviewer 3 Report

This paper is unpublishable as it stands, mainly because of the atrocious English. Even the title has a spelling mistake! It is possible to initiate a Mode I fatigue crack under compression-compression loading (Stage I) but it will not grow (Stage II) unless there is a tensile residual stress to counteract the compression. The only exception to this is a shear mode crack (Mode II and/or Mode III) which grows for a short distance under the action of cyclic shear, not tensile, stress. I note that the experimental testing is under wholly tensile loading, but the numerical simulation appears to be under wholly compressive loading (there would be no actual fatigue crack growth in this case). The description of fatigue in the Introduction is erroneous. For example, I quote: “Vacancies generated when an atom or an ion is missing from its regular crystallographic site and new equilibriums is built. Under repeated loading, the new dynamic equilibriums lost to generate the next balances, and then the fatigue damage initiates and accumulates in a natural way”. Fatigue cracks initiate at stress-raisers, either surfaces or internally at inclusions. Stage I (initiation) occurs through dislocations moving back and forwards on slip planes within individual grains. Stage II (propagation) occurs by means of separation of material at the crack tip in response to a tensile opening stress.

Round 2

Reviewer 2 Report

The authors have not followed my suggestions and comments. This last version of the paper shows the same limits of the first one.

Reviewer 3 Report

I still have problems with this paper. While it is possible to get fatigue initiation under compression-compression loading, subsequent propagation is only possible if there is tensile (residual) stress or the fatigue crack grows in a shear mode and subsequently gives off a tensile (Mode I) branch crack. It is stated that the laboratory fatigue tests were performed under compression-compression loading but that the stress ratio R was 0.1 (tension-tension loading). I would recommend that the authors read the excellent book Metal Fatigue by Les Pook, Springer, 2007. If the paper is to be published then at least correct the spelling mistake in the title (Sate -> State).